# ExlA Pore-Forming Toxin: Localization at the Bacterial Membrane, Regulation of Secretion by Cyclic-Di-GMP, and Detection In Vivo

**DOI:** 10.3390/toxins13090645

**Published:** 2021-09-11

**Authors:** Vincent Deruelle, Alice Berry, Stéphanie Bouillot, Viviana Job, Antoine P. Maillard, Sylvie Elsen, Philippe Huber

**Affiliations:** 1Unité de Biologie Cellulaire et Infection, Université Grenoble-Alpes, CEA, INSERM, CNRS, 38054 Grenoble, France; vincent.deruelle@pasteur.fr (V.D.); berry.alice.fr@gmail.com (A.B.); stephanie.bouillot@cea.fr (S.B.); viviana.job@cea.fr (V.J.); antoine.maillard@cea.fr (A.P.M.); Sylvie.elsen@cea.fr (S.E.); 2Unité de Biochimie des Interactions Macromoléculaires, Département de Biologie Structurale et Chimie, CNRS UMR 3528, Institut Pasteur, 75015 Paris, France; 3Institut de Biologie Structurale (IBS), Université Grenoble-Alpes, CNRS, CEA, 38044 Grenoble, France; 4Center for Immunology of Viral, Auto-Immune, Hematological and Bacterial Diseases (IMVA-HB/IDMIT), Université Paris-Saclay, Inserm, CEA, 92265 Fontenay-aux-Roses, France

**Keywords:** bacterial pore-forming toxin, two-partner secretion, c-di-GMP, *Pseudomonas aeruginosa*, toxin dosage

## Abstract

ExlA is a highly virulent pore-forming toxin that has been recently discovered in outlier strains from *Pseudomonas aeruginosa*. ExlA is part of a two-partner secretion system, in which ExlA is the secreted passenger protein and ExlB the transporter embedded in the bacterial outer membrane. In previous work, we observed that ExlA toxicity in a host cell was contact-dependent. Here, we show that ExlA accumulates at specific points of the outer membrane, is likely entrapped within ExlB pore, and is pointing outside. We further demonstrate that ExlA is maintained at the membrane in conditions where the intracellular content of second messenger cyclic-di-GMP is high; lowering c-di-GMP levels enhances ExlB-dependent ExlA secretion. In addition, we set up an ELISA to detect ExlA, and we show that ExlA is poorly secreted in liquid culture, while it is highly detectable in broncho-alveolar lavage fluids of mice infected with an *exlA*+ strain. We conclude that ExlA translocation is halted at mid-length in the outer membrane and its secretion is regulated by c-di-GMP. In addition, we developed an immunological test able to quantify ExlA in biological samples.

## 1. Introduction

*Pseudomonas aeruginosa* is an opportunistic pathogen causing acute infections, notably ventilator-associated pneumonia, as well as chronic—often fatal—infectious diseases in cystic fibrosis patients [1,2]. A high proportion of *P. aeruginosa* clinical strains are multiresistant to antibiotics, which leaves little options for clinical management of patients infected with these strains.

Several virulence factors have been identified that are differentially expressed in *P. aeruginosa* strains. The most potent virulence factor is a type 3 secretion system (T3SS) injecting toxins in the cytoplasm of host cells [3]. Four T3SS effectors have been identified—ExoU, ExoS, ExoT and ExoY—that trigger major alterations in the host cell and eventually lead to cell death. Recently, genetic outliers of *P. aeruginosa* have been isolated from infected patients [4,5,6,7]; they do not possess the genes encoding the T3SS, nor those for the T3SS toxins, but instead they contain a pathogenicity locus of two genes, named *exlA* and *exlB*, encoding a two-partner secretion (TPS) system. The *exlA* gene product, called ExlA or exolysin, is the passenger protein of the TPS, and the *exlB* gene product, ExlB, is the transporter protein, located in the bacterial outer membrane, that allows ExlA secretion in the extracellular milieu. ExlA is a 172-kDa protein harboring (i) at the N-terminal end, a signal peptide (SP) followed by a TPS domain allowing interaction with its transporter ExlB, (ii) in the central part, a shaft consisting of a type-2 filamentous haemagglutinin adhesin (FHA2) domain, and (iii) a C-terminal domain that behaves like a molten globule in solution and has no characterized homolog [8,9] (Appendix A). Interestingly, ExlA-like toxins have been identified in other *Pseudomonas* species (*P. putida*, *P. entomophila* and *P. protegens*) [10,11]. The ExlA-ExlB TPS is structurally related to TPSs present in other bacterial pathogens and exhibiting cytolytic activities, including ShlA-ShlB in *Serratia marcescens*, HpmA-HpmB in *Proteus mirabilis*, EthA-EthB in *Edwardsiella tarda* and HhdA-HhdB in *Haemophilus ducreyi* [12].

ExlA is a pore-forming toxin that acts as a cytolysin for most human cell types tested, i.e., epithelial, endothelial, fibroblastic, myelocytic and lymphoblastic cells, but it is poorly toxic for erythrocytes [6]. Furthermore, the cytotoxicity scores of *exlA*+ strains were correlated with the levels of secreted toxin [6], confirming that ExlA is the main virulence factor in these strains.

Upon bacteria–host cell interaction, ExlA rapidly inserts into lipid rafts of host cell plasma membrane [9]. ExlA insertion within the host membrane induces the formation—by unknown mechanisms—of a pore 1.6 nm in diameter, triggering both Ca^2+^ influx and K^+^ efflux [8,10,13]. In epithelial cells, Ca^2+^ influx leads to rapid cell-cell junction disruption via the activation of ADAM10 metalloprotease, followed by cell osmolysis [13]. In macrophages, K^+^ efflux induces the assembly of the NLRP3 inflammasome, causing cell death by pyroptosis [10].

Transcription of the *exlBA* operon is activated by the CyaB-cAMP/Vfr pathway, also known to regulate the T3SS, and is repressed by ErfA, a recently discovered transcription factor [14,15]. The strains secreting the largest amounts of ExlA were highly toxic in a mouse model of acute pneumonia and led to rapid mouse death [6,16,17]. In these in-vivo models, the bacteria induced major ExlA-dependent lesions of the alveolo-capillary barrier, provoking lung hemorrhages and bacterial dissemination in other organs [16,17].

*exlA*+ strains have been isolated in patients with different types of infectious disease [6,18,19] and are widespread in the environment [6,20]. In most reported studies on *exlA*+ outliers, strain identification has been performed by genetic analysis of the isolates, but did not measure the amounts of secreted ExlA, a crucial parameter to determine strain virulence. Furthermore, the secretion mechanism of the TPS pore-forming toxins remains elusive. Previous data suggest that ExlA is not a diffusible toxin, but requires contact with the host cell for its delivery [8]; a feature that has been recently reported for another TPS passenger protein, CdiA, which remains entrapped inside the transporter protein until the external part of the passenger interacts with a receptor located on the target cell [21].

Here, we employed various approaches to determine the amounts of secreted vs membrane-bound ExlA. We confirm that ExlA is poorly secreted in the culture medium and we show that bacteria harbor spots of immunolabeled ExlA at their membrane, which are dependent on the presence of ExlB. We further demonstrate that ExlA secretion via ExlB is regulated by the second messenger c-di-GMP. Finally, ExlA was detected and quantified by ELISA in broncho-alveolar lavage (BAL) fluids of infected mice, and their ExlA contents were correlated with the levels of disease severity markers.

## 2. Results

### 2.1. ExlA Quantification in Bacterial Secretomes

ExlA is difficult to quantify in bacterial secretomes, because of its instability and low abundance. Its detection by Western blot requires a preliminary precipitation step. To measure ExlA release in bacterial cultures directly and more precisely, we set up a sandwich ELISA using polyclonal and monoclonal antibodies against ExlA (see Figure 1A,B and Appendix A and Section 4). Supernatants of cultures, grown in presence of EGTA to prevent proteolysis, were collected, supplemented with an anti-protease cocktail and frozen. IHMA, an *exlA*+ strain, IHMAΔ*exlBA*, lacking ExlA and ExlB, as well as IHMAΔ*erfA*, lacking the *exlBA*-locus repressor ErfA and producing high levels of ExlA, were used in this experiment. PAO1, a *T3SS*+/ *exlA*- strain, was also used in this assay as negative control (see Table 1 for strain description). Low, but significant, ExlA concentrations were detected in the secretomes of IHMA, and background signals were recorded for PAO1 and IHMAΔ*exlBA* secretomes (Figure 1C). This feature confirms that ExlA is poorly secreted from wild-type *exlA*+ strains in these conditions, although IHMA exhibited high ExlA-dependent toxicity in cellular and in-vivo models of infection [4,6,16,17]. Conversely, high ExlA levels were measured in IHMAΔ*erfA* secretomes, indicating that the minimal ExlA contents detected in the IHMA secretome were due to a weak secretion, rather than to an artifact of the detection method.

### 2.2. ExlA Localization at the Bacterial Membrane

To shed light on ExlA secretion pathway, we immunolabeled ExlA in immobilized bacteria. Different ExlA antibodies were tested, but only the 7G10 antibody (Appendix A) generated specific labeling, consisting of spots located at the bacterial membrane of IHMA, while only background signals were obtained with strains IHMAΔ*exlBA* or IHMAΔ*exlB* (Figure 2A,B). Approximately 0.5% of bacteria harbored at least one spot in IHMA (Figure 2C). Immunolabeling of a strain devoid of ErfA (IHMAΔ*erfA*), a highly active repressor of the *exlBA* operon [15], increased the proportion of bacteria with spots (up to 10%) and their intensity (Figure 2A–C). The spots were localized in any regions of the bacterial membrane (Figure 2D). When Polymyxin B was added to permeabilize the outer membrane, the immunolabeling still consisted of spots for IHMA and IHMAΔ*erfA*, with an increased number for the latter (Figure 2A,C). Large patches were observed in the periphery of polymyxin B-treated IHMAΔ*exlB* bacteria, suggesting that ExlA spilled out of the cell (Figure 2A). Interestingly, the number of spots or patches is similar for permeabilized IHMA and IHMAΔ*exlB*, indicating that *exlB* mutation does not impact the proportion of ExlA+ cells.

Taken together, these results indicate that ExlA accumulates at specific locations of the bacterial membrane, likely associated with ExlB, instead of or in parallel with its release in the extracellular medium. Interestingly in IHMAΔ*erfA*, the signals were much more intense than those observed with IHMA, suggesting that ExlB pores, associated with ExlA, clusterize at the bacterial membrane. Thus, lifting transcriptional repression has two effects: not only the secretion rate is enhanced (Figure 1C) and [15], but also more ExlA is located at the membrane.

### 2.3. Cyclic-di-GMP Controls ExlA Secretion

Having shown that ExlA is present in specific locations of the bacterial membranes, we sought the mechanisms controlling ExlA release from the membrane to the extracellular milieu.

We noticed that, in liquid culture, IHMA bacteria tend to form small aggregates and are hyperadherent to plasticware (not shown), a phenotype known to be activated by the second messenger c-di-GMP [26,27]. Furthermore, IHMA exhibits a reduced swarming motility [6], in agreement with a high cellular content of c-di-GMP [28,29]. Therefore, we analyzed the intracellular c-di-GMP levels of IHMA, compared to PAO1, using the fluorescent reporter *PcdrA-gfp*, where the *cdrA* promoter -activated by c-di-GMP- is fused to the *gfp* gene [26]. The fusion, carried on a mini-Tn7 transposon, is stably integrated at a neutral *attTn7* site in the bacterial chromosome. The fluorescence intensities increased in both strains during bacterial growth, but the signals were significantly higher for IHMA (Figure 3A,B). This feature suggests that IHMA contains higher c-di-GMP amounts than PAO1.

We reasoned that if c-di-GMP regulates ExlA synthesis or secretion, manipulating c-di-GMP concentrations should alter toxin release. To modulate c-di-GMP levels in IHMA, we used integrative vectors derived from plasmids that either decrease its content by expression of the phosphodiesterase PA2133 or increase c-di-GMP synthesis by expression of the diguanylate synthase WspR^R242A^ (WspR*) [23]. As expected, expression of PA2133 (strains “c-di-GMP −”) significantly decreased its expression, as monitored by *PcdrA*-GFP fluorescence (Figure 3C,D). Expression of WspR* (strains “c-di-GMP +”) did not significantly alter c-di-GMP expression, indicating that WspR* overexpression has no effect on c-di-GMP concentrations. We used these systems of c-di-GMP manipulation to examine the role of this messenger in ExlA secretion by Western blot (Figure 3E,F). The synthesis of PA2133 in IHMA increased the presence of ExlA in secretomes and decreased its level in bacterial extracts, compared to wild type IHMA. Expression of WspR* did not alter ExlA contents in secretomes and bacterial extracts, in line with the negligible effect of WspR* expression on bacterial c-di-GMP contents (Figure 3C,D). To further assess the role of c-di-GMP in ExlA secretion regulation, we measured ExlA in secretomes by ELISA using the same approach. As ExlA concentrations are very low in IHMA secretomes, we inserted *PA2133* and *WspR** in the IHMAΔ*erfA* background. The results confirmed that lowering c-di-GMP increases ExlA concentration in secretomes (Figure 3G).

Cyclic-di-GMP can modulate the transcriptional activity of several genes, such as those coding for the TPS CdrAB of *P. aeruginosa* [30]. We tested this possibility for ExlAB by using the reporter system *exlA-lacZ*, in which the *lacZ* gene is under the control of the *exlBA* promoter [14]. The β-galactosidase activities of IHMA, IHMA-c-di-GMP + and IHMA-c-di-GMP—strains were not significantly different (Figure 3H), indicating that c-di-GMP does not regulate *exlBA* transcription. To assess whether c-di-GMP affects ExlA intracellular stability, we used the IHMAΔ*exlB* strain, which does not secrete ExlA [8]. Manipulation of c-di-GMP levels in IHMAΔ*exlB* did not modify intracellular ExlA contents (Figure 3I), as opposed to what was observed in IHMA or IHMAΔ*erfA* (Figure 3E–G).

Taken together, these results indicate that high c-di-GMP levels in IHMA block ExlB-dependent ExlA release into the extracellular milieu, without affecting ExlA expression or stability.

### 2.4. ExlA Is Retained at the Outer Membrane

To regulate ExlA export, c-di-GMP may act on its transportation across the internal or external membranes. ExlA is predicted to be transported into the periplasm by the translocation system Sec. To determine whether ExlA accumulates in the cytosol or in the periplasm, bacterial extracts from IHMAΔ*erfA*-c-di-GMP + and IHMAΔ*erfA*-c-di-GMP—were fractionated in order to isolate cytosol, periplasm and membranes. Total bacterial extracts and secretomes were also prepared. The strain IHMAΔ*erfA* was used instead of IHMA to enrich the fractions in ExlA contents. The different fractions were analyzed by Western blot in order to detect ExlA, as well as DsbA and Opr86, as markers for periplasm and membranes, respectively. The amounts of DsbA in the periplasm were similar in the two conditions (c-di-GMP + vs c-di-GMP −), showing that the Sec machinery is not affected by c-di-GMP levels (Figure 4A,B). Furthermore, minimal ExlA signals were detected in the cytosol and periplasm in both conditions, while strong signals were observed in the membranes. These findings are in agreement with an accumulation of ExlA at the outer membrane, likely inside ExlB pores. ExlA amount was increased in the secretome of the c-di-GMP − strain, indicating that c-di-GMP also affects ExlA release in IHMAΔ*erfA* background.

To definitely prove that ExlA is inserted in the outer membrane, we subjected intact IHMA bacteria to proteinase K-dependent proteolysis (Figure 4C,D). We observed a dose-dependent decrease of ExlA signals, while the periplasmic protein DsbA was not degraded, indicating that the protease had no access to this compartment. Thus, a part of ExlA protein points out of the bacterial membrane and is accessible to the protease. A similar experiment was performed with IHMAΔ*exlB*, for which no ExlA decrease was observed, indicating that ExlB is required for ExlA positioning in the outer membrane with an accessible domain outside the bacterial membrane.

Altogether, our findings show that ExlA is entrapped in ExlB at the bacterial outer membrane. This location is favored by high c-di-GMP levels, while c-di-GMP downregulation increases ExlA secretion.

### 2.5. ExlA Detection in Broncho-Alveolar Lavage Fluids of Infected Mice

One of the main issues of this study was the detection of ExlA in biological fluids of infected individuals, and notably in broncho-alveolar lavage (BAL) fluids. We thus performed BALs in infected mice to measure ExlA concentration in these fluids. In this experiment, mouse lungs were infected by inhalation of a bacterial suspension. Lavages, with PBS supplemented with an antiprotease cocktail, were performed at 18 h post-infection in euthanized mice. Mice were infected either with IHMA or PAO1, or were mock-infected with PBS. High ExlA concentrations, ranging from 20 to 60 ng/mL, were measured in the supernatant of centrifuged BAL fluids from IHMA-infected mice, while only background levels were found in PAO1 and PBS samples (Figure 5A).

To see whether the variations of ExlA contents between samples of IHMA-infected mice were correlated with the degree of infection, we measured markers that were previously established as hallmarks of pathogenicity of ExlA-positive strains, i.e., protein, IL-6 and hemoglobin levels present in BAL fluids [4,16,17]. The measured concentrations of each of these markers correlated well with ExlA levels in these samples (Figure 5B–D), suggesting that ExlA levels are directly linked to disease severity.

## 3. Discussion

Among the extensive superfamily of bacterial pore-forming toxins, the mode of action of members of the TPS family, which includes ExlA from *P. aeruginosa* and ShlA from *S. marcescens*, remains the most elusive. The regulation of their secretion is largely unknown and no host receptor has so far been identified, as opposed to most studied pore-forming toxins [31]. In addition, these toxins are remarkably difficult to detect in and to isolate from bacterial secretomes or biological fluids.

The detection of ExlA or ShlA in bacterial secretomes requires the addition of anti-proteases and a concentration step by TCA-precipitation prior to Western blot analysis [4,6,32]. Here, low ExlA amounts were measured by ELISA in bacterial secretomes from wild-type IHMA grown in liquid culture, confirming its low abundance in this conditioned medium (Figure 1). Conversely, when ExlA was overproduced in IHMAΔ*erfA* strain, high amounts were measured. This feature suggests that low secretion, rather than proteolytic degradation, is responsible for its low abundance in bacterial secretomes. In addition, we showed by immunofluorescence (Figure 2) and cellular fractionation (Figure 4A,B) that ExlA accumulates at the bacterial membrane. The fact that bacterial ExlA is sensitive to proteinase K degradation (Figure 4C,D) indicates that ExlA is localized at the outer membrane where at least part of the protein points out of the bacterial surface. No ExlA spot was observed in unpermeabilized or permeabilized *exlB* mutant strain (Figure 2). Rather, ExlA spilled out of the bacteria in permeabilized IHMAΔ*exlB*, as opposed to permeabilized IHMA, for which only ExlA spots were observed. This result further suggest that ExlA and ExlB are associated and that ExlB anchors ExlA at the outer membrane.

In a TPS system, secretion of the passenger protein through outer membrane-embedded transporter cannot be directly supported by energy sources, like ATP or an electrochemical gradient [12]. This feature suggests that the secretion process may be achieved in successive steps separated by low energy barriers. In the case of contact-dependent inhibition (CDI) toxins, belonging to the TPS family, it has been recently shown that passenger translocation across the outer membrane stalls at mid-length and that secretion through its TPS transporter resumes upon interaction with a receptor on the target cell [21]. An intermediate state, in which the passenger is entrapped into the transporter, has also been reported for other TPS proteins, like HMW1 [33] and CdrA [34], where another protein partner is needed to release the TPS-A substrate.

Although no specific receptor is known for pore-forming toxins related to ExlA, the absence of toxicity of filtered secretomes and the lower toxicity of isogenic strains devoid of type 4 pili, which are known to promote bacterium-host cell interaction [8], can both be explained by ExlA secretion stalling within ExlB at the outer membrane. From this perspective, it can be hypothesized that the active toxin is membrane-bound and not secreted.

As postulated by Nash and Cotter [35], using *Bordetella* filamentous hemagglutinin as model, the TPS translocation pathway may represent a mechanism of regulated toxin delivery. Here, we show that ExlA release is regulated by intracellular c-di-GMP. Lowering c-di-GMP levels induced ExlA release from the membrane into the extracellular milieu (Figure 3E–G and Figure 4A,B). Thus, bacteria can control ExlA-dependent toxicity not only at the transcriptional level via Vfr and ErfA, but also at the secretion step via c-di-GMP levels. Cyclic-di-GMP is normally only present in the cytosol, not in the periplasm, thus indicating that the regulation mechanism is indirect. Therefore, a transduction pathway must convey the c-di-GMP signal from the cytosol to ExlA and ExlB located in the outer membrane. Such a mechanism of transduction has been described for *P. aeruginosa’s* TPS CdrA-CdrB, for which CdrA release is also inhibited by cytosolic c-di-GMP. In this system, CdrA threading through CdrB is stalled due to a C-terminal hook domain shaped by an internal disulfide bridge. CdrA secretion resumes when the periplasmic protease LapG cleaves off the hook domain, hence allowing CdrA release. In this regulation system, c-di-GMP neutralizes LapG via the activation of its inner membrane receptor LapD that sequesters LapG at the inner membrane [34]. However, this mechanism cannot be applied to ExlA-ExlB, as only one Cys residue is found in the ExlA sequence, located in the peptide signal. Furthermore, inactivation of *lapD* did not alter ExlA secretion (data not shown), in agreement with the absence in ExlA sequence of the protease cleavage motif. More work is needed to fully decipher the c-di-GMP/ ExlA regulation axis.

Our main conclusions and hypotheses on ExlA secretion regulation are shown in Figure 6. Two mechanisms of TPS-A translocation currently appear in the literature, that apply to different biological models. For proteins like HMW1A and CdrA, a linear model has been established where the N-terminal end is secreted first through the pore and points out of the membrane, whereas for CdiA—and possibly FHA—a hairpin intermediate has been described, where the central part of TPS-A protrudes [12,21]. As the conformation of ExlA bound to ExlB is unknown, the linear model is presented in Figure 6 for the sake of simplicity.

TCA precipitation followed by Western blot is not relevant for BALs from infected mice, because of the high protein concentrations present in these fluids. ExlA was not detected in BALs by LC-MS/MS approaches either for the same reason (not shown). The ELISA developed here is the first method that could detect ExlA and quantify its concentration in BAL fluids. Importantly, the presence of antiproteases in the buffer used for the lavages is required for optimal ExlA detection in the assay. Why ExlA is highly secreted in vivo and poorly in vitro remains an open question. No secretion enhancement was observed when IHMA was co-incubated with human cells (not shown), suggesting that the pulmonary micro-environment is required for secretion enhancement, which may involve the downregulation of ErfA repression and/or the decrease in bacterial c-di-GMP content. Alternatively, ExlA may be present in cellular debris (from bacteria or host cells) that were not eliminated by the centrifugation step.

The ExlA levels measured in BALs from individual mice correlated with disease severity, as monitored by dosage of infection markers. Implementation of this assay into the clinic could be useful to identify *exlA*+ strains in infected patients. Although no specific treatments have been developed to combat infections with bacteria producing this highly potent toxin, ExlA detection may provide a decision support for symptomatic treatments (i.e., anti-inflammatory or antithrombotic drugs) or for patient isolation to prevent transmission.

## 4. Materials and Methods

### 4.1. Ethics Statement

All protocols in this study were conducted in strict compliance with the French guidelines for the care and use of laboratory animals. The protocols for mouse infection were approved by institutional animal research committees (CETEA#44, project number 02636.01) and the French Ministry for Research on 19 May 2016.

### 4.2. E. Coli and P. Aeruginosa Strains and Culture

The strains used in this study are described in Table 1. Bacteria were grown in liquid LB medium at 37 °C with agitation, except for ExlA purification where minimal medium M9 and 30 °C were selected instead.

### 4.3. Cloning

Two compatible plasmids have been used to coproduce ExlB (pACYC-Duet) and ExlA (pET28) in *E. coli*. The sequence of *exlB* was not only codon-optimized but also modified such as the mature peptidic sequence displayed an 8-histidine tag at its N-terminus and was fused downstream *E. coli* OmpA signal peptide. The *exlB* chimeric gene was subcloned into pACYC-Duet (Novagen) using NcoI and AvrII restriction sites, so that a single T7 promoter is present in the recombinant plasmid. The mature sequence of *exlA* was codon-optimized for expression in *E. coli* [8] and cloned such as to be produced downstream of the *D. dadantii* PelB signal peptide and to display a 9-histidine tag at its C-terminus. The *exlA* chimeric gene was assembled by ligating PCR-amplified *pelB* signal sequence and *exlA* mature sequence into pET28 (Novagen). *pelB* was amplified from pET9-ss-10His-tev-LIC [36] with the primers SD-pET9-Xba-5 [5-ggagactacaacggtttcccTCTAGAaataattttg-3] and pelBss-Nco-3 [5-TTAGCCATGGCCATCGCCG-3]. The His-tagged *exlA* mature sequence was amplified from pET28a-exlA_noSP_ using mature-exlA-Nco-5 [5-GCACCATGGGCGGTCTGGAAGCCG-3] and exlA-9His-stop-Nhe-3 [5-GCTGCTAGCTCAATGGTGATGGTGATGGTGATGGTGATGCCCATCCGCTTTCTGTTCAATACCCGC-3] as the primer pair. Nine *exlA* variants have been derived from pET28-pelB-exlA-9His. The pET28-pelB-exlA347, 843, 1224 variants were, respectively, obtained by NotI, SalI, SacI restriction digest and re-ligation of pET28-pelB-exlA-9His as these sites are present both in the optimized *exlA* sequence and in the pET28 cloning site downstream *pelB*-*exlA-9His*. pET28-pelB-exlA-1356 (ΔCter) was obtained by subcloning the SacI-XhoI fragment of pET28-exlA_noSP_-ΔCter [8] into pET28-pelB-exlA-9his. pET28-pelB-exlA278 was made by PCR cloning using SD-pET9-Xba-5 and exlAsam-P278-6His-Nhe-3 [5-gctGCTAGCtcaGTGATGGTGATGGTGATGcggaccaacacgcacacccg-3]. pET28-pelB-exlA1465 and pET28-pelB-exlA1526 were derived by cloning the PCR fragments obtained with exlAsam-seq3583-GSV1193 [5-catcgaaaaacggcagtg-3] and either exlAsam-4395-Xho-3 [5-cctCTCGAGCTAgtagccgcgcgcacgatccagtttggacg-3] or exlAsam-4610-Xho-3 [5-cctCTCGAGCtaattacgcgacacatgtgaagcgtcggggtc-3] into the SacI-XhoI linearized pET28-pelB-exlA-9His.

The integrative pSW196-wspR* and pSW196-PA2133 vectors derived from pBBR1MCS4-R246A-wspR and pBBR1MCS4-PA2133, respectively, by transferring the *Xba*I-*Sac*I fragments encoding the active guanylate cyclase WspR^R246A^ and the phosphodiesterase PA2133 into the pSW196 vector cut with the same enzymes. Thus, the genes were placed under control of arabinose-inducible P*BAD* promoter. The vectors were transferred into *P. aeruginosa* IHMA by triparental mating using pRK600 as a helper plasmid, and the integration events at *attB* site were selected on LB plates containing 25 µg/mL Irgasan and 75 µg/mL tetracycline. The pTn7 CdrA-gfp(ASV)^c^ suicide delivery vector was introduced into IHMA along with pUX-BF13 that encodes the Tn7 transposase by tetraparental mating. The integration of mini-Tn7 carrying the *PcdrA-gfp* fusion at the *attTn7* site was selected on LB plates containing 25 µg/mL Irgasan and 75 µg/mL gentamicin.

### 4.4. Production and Purification of ExlA and ExlA Truncated Derivatives

*E. coli* strains coproducing ExlB and ExlA derivatives were grown at 30 °C in M9 minimal medium supplemented with 0.2% glucose, 0.2% casaminoacids and 1 mM thiamine.

Antigen production was induced with 0.3 mM IPTG for two hours until cultures reached OD_600 nm_ 0.8. After centrifugation in a JLA-9.1 rotor (Beckman) at 6500 rpm and 8 °C for 25 min, both the supernatant and the pellet were processed as follows. The pellet was resuspended in 3 mL extraction buffer (20 mM HEPES pH 8.0, 160 mM NaCl, 6 M urea, pH 8.0) and incubated at room temperature for 10 min. The suspension was cleared by centrifugation at 18,000× *g* for 5 min and 8 °C. Urea was then diluted to 4 M to yield the inoculation antigen (1 g/L determined by Bradford assay).

For the antibody screening by direct ELISA, the bacterial supernatant was used instead of the purified protein from the bacterial pellet. The supernatant was vacuum-filtered through an 0.2 µm filter (Millipore). Mock samples from ExlA-negative bacterial supernatants were also prepared, so as to eliminate *E. coli* cross-reacting hybridomas. The antibody binding properties of ExlA derivatives were assessed using urea extracts prepared from 18,000× *g* pellets of minicultures.

### 4.5. Production of Monoclonal Antibodies

Antibodies were produced in the mouse by BIOTEM (Apprieu, France), using established procedures. Briefly, purified ExlA (see below) was injected together with Freund’s adjuvant. Hybridomas were generated and subcloned until 100% of subclones produced ExlA antibodies. Antibodies from conditioned medium were purified by protein A chromatography.

### 4.6. Antibody Mapping on ExlA Sequence

To localize the antibody binding sites, several ExlA truncated variants were produced (Appendix A). Antibody immunoreactivity toward seven deletion fragments and full-length ExlA protein was first examined by Western blot analysis (Appendix A). The 5H6 monoclonal antibody binds to the N-terminal region (between residues 347 and 843). The 7G10 monoclonal antibody induced no signal by this technique, indicating that this antibody is not reactive with denatured ExlA, and thus suggesting that it recognizes a conformational epitope. To overcome this lack of immunoreactivity, the 7G10 antibody was tested in dot-blot experiments, which better preserves protein conformation (Appendix A). With this technique, the 7G10 displayed high immunoreactivity with fragments 843 and longer; a weak signal was also observed with the 347 fragment. This feature suggests that the main part of the 7G10 epitope is located in the 347–843 fragment, and that the 278–347 sequence may also contain part of the recognition sequence. The results of monoclonal antibody mapping are summarized in Appendix A, together with the positions of previously produced rabbit polyclonal antibodies against ExlA Cter (“Cter”) and ExlA Nter (“ΔCter”) fragments [14].

### 4.7. ExlA ELISA

The capture antibody (ΔCter polyclonal antibody) was used at 0.1 μg/mL in coating buffer (50 mM sodium carbonate pH 9.6) and 100 µL per well of the diluted antibody were distributed in a 96-well plate (Nunc-MaxiSorp, ThermoFisher, 456537). The plate was sealed and incubated overnight at 4 °C without shaking. The buffer was discarded and the wells were washed four times with 300 µL of washing buffer (0.05% Tween20 in PBS). The wells were subsequently saturated with 200 µL of blocking buffer (BSA 1% in PBS) for 2 h at room temperature with shaking (500 rpm). After four washes, 100 µL of samples or standard were loaded in triplicate and incubated for 2 h at room temperature with shaking. After four washes, 100 µL of detection antibody (5H6 mouse monoclonal antibody; 2.66 µg/mL in blocking buffer) were added and incubated for 2 h at room temperature with shaking. After four washes, 100 µL of secondary antibody (anti-mouse HRP antibody, 2.2 µg/mL in blocking buffer) were added and incubated for 1 h at room temperature with shaking. After four washes, 100 µL of HRP substrate (TMB; Sigma-Aldrich T0440) were added, followed by a 10-min incubation at room temperature with shaking in the dark. The reaction was stopped by adding 100 µL of 2N H_2_SO_4_. Plates were read in a spectrophotometer (SAFAS) at 490 nm.

Samples (secretomes and BALs) were assayed undiluted or diluted in blocking buffer, and purified ExlA was used as standard.

### 4.8. Production of Bacterial Secretomes

Bacteria were grown in LB supplemented with 5 mM EGTA/20 mM MgCl_2_, until they reached DO_600_ = 2.0. Bacteria were eliminated by two centrifugations at 4000× *g* and 10,000× *g*, and supernatants were used as bacterial secretomes.

### 4.9. Immunofluorescence of Bacteria

We used four *P. aeruginosa* strains: IHMA, IHMA *exlB*-mut (IHMAΔ*exlB*), IHMAΔ*exlBA* and IHMAΔ*erfA*. When bacterial cultures reached the exponential phase (OD_600_ = 1.0), chloramphenicol (500 µg/mL) was added to block protein synthesis for 30 min at room temperature (RT). Then, 3 mL of bacteria were fixed with paraformaldehyde (PFA) 4% in 250 mM HEPES pH 7.4 and incubated overnight at 4 °C before immunolabeling. Briefly, the bacteria were centrifuged at 3000 rpm for 15 min and washed twice with PBS with 1 mM MgCl_2_ (PBS/Mg) to remove PFA. Then, each samples were separated in two: 2 mL were permeabilized with 100 µg/mL of Polymyxin B for 5 min at RT, then washed in PBS/Mg; 1 mL was not permeabilized. Both preparations were incubated 30 min in PBS/Mg containing 0.5% BSA (BS), spun down, and then incubated 30 min at RT in BS containing the anti-ExlA purified antibody (7G10, diluted 1:250, 22 µg/mL final concentration). Finally, bacteria were washed three times with PBS/Mg and incubated 30 min with BS containing the anti-mouse antibody coupled to Alexa Fluor^®^A488 (Lifetechnologies, #A21202, diluted 1:500, 4 µg/mL final concentration), vital Hoechst (diluted 1:500 at 20 µg/mL final concentration) and when indicated with FM^®^4–64 (Molecular probe; #T3166, diluted 1:500 at 2 µM final concentration) to stain DNA and membranes, respectively. After three final washing steps in PBS/Mg, bacteria were mounted on 1.5% agar pads and visualized under an Olympus microscope with a 100 × objective.

Images analysis was performed using the MicrobeJ plugin of ImageJ [37]. For ExlA spot “maxima” detection, we used the “foci” setting with a tolerance of 130 and filtered the intensity from 200-max to remove noise. The same settings were used for all images. Each maximum detected was exclusively associated with only one bacterium, thanks to MicrobeJ association “outside/inside” settings, with a maximum distance of 0.5 µm for “outside” and 0.1 µm for “inside” association. The number of bacteria with ExlA spots was calculated from 10–11 independent images for each sample.

### 4.10. Fluorescence of GFP Bacteria

The fluorescence of bacteria containing the *PcdrA-gfp* reporter fusion was analyzed with the fluorospectrophotometer Fluoroskan Ascent (Labsystems) set at 37 °C. Bacteria at OD_600_ 0.05 were distributed (100 µL per well) in a dark-well microplate (Greiner). Fluorescence (Ex 485 nm, Em 527 nm) was measured every 15 min for 16 h. In parallel, OD_585_ was recorded to monitor bacterial growth.

### 4.11. Total Protein Extracts from Secretomes and Bacteria

For the secretomes, proteins from the culture supernatants were precipitated by addition of sodium deoxycholate (0.02% final concentration) and trichloroacetic acid (TCA, 10% final concentration), and incubation at 0 °C for 2 h. Proteins were pelleted at 15,000× *g* for 15 min, and resuspended in loading buffer at 1:100 of the culture volume. For IHMAΔ*erfA*, no concentration of the secretomes is required. For total bacterial extracts, the bacterial pellets were solubilized in 1:10 of the culture volume.

### 4.12. Cellular Fractionation

To isolate proteins from the different cellular compartments, 30 mL of bacterial culture were centrifuged at 600 rpm for 15 min. The pellets were resuspended in 1 mL of Buffer M (20 mM Tris-HCl, 200 mM MgCl_2_, pH 8.0, with antiprotease cocktail from Roche) supplemented with lysozyme (0.5 mg/mL), and incubated 30 min at 4 °C with rotation. The suspension was then centrifuged at 11,000 rpm for 15 min. The supernatants corresponded to the periplasm and were conserved. The pellets were washed with Buffer B (20 mM Tris-HCl, 20% saccharose, pH 8.0, with antiprotease cocktail) followed by centrifugation. The pellets were resuspended in Buffer M and sonicated (5 min with cycles of 10 s ON and 10 s OFF, with 40% amplitude at 4 °C). Cell debris were pelleted by centrifugation at 6000 rpm for 15 min, and the supernatants were ultracentrifuged at 200,000× *g* for 45 min. The supernatants correspond to the cytosolic fraction and were conserved. After washing the pellets with Buffer A (20 mM Tris-HCl, 20 mM MgCl_2_, pH 8.0, with antiprotease cocktail) and recentrifugation, they were resuspended in 500 µL of Buffer B and sonicated. This latter fraction corresponded to total membranes.

### 4.13. Proteinase K Treatment

The methods is adapted from Faure et al. 2014 [38] with modifications. Briefly, 2 mL of bacterial culture at OD_600_ were centrifuged for 10 min at 8000 rpm and bacteria were resuspended in 1 mL of 20 mM Tris-HCl, 10 mM MgCl_2_, pH 8.0. The suspension was incubated with proteinase K (concentration range 3–30 ng/mL) in ice for 5 min. Bacteria were then centrifuged for 5 min at 2375 g and the pellet was rapidly resuspended in denaturing loading buffer followed by heating at 100 °C for 10 min.

### 4.14. Western Blot

Protein extracts were analyzed by electrophoresis and transferred onto PVDF membranes by standard procedures. Membranes were incubated with the following antibodies: Cter and ΔCter ExlA antibodies, FliC [14], EFTu (Hycult Biotech, HM6010), DsbA and Opr86 [39].

### 4.15. β-Galactosidase Activity

β-galactosidase activity was assayed as described [40], with technical details reported in Thibault et al. 2009 [41].

### 4.16. Mouse Pulmonary Infection

Pathogen-free C57Bl/6 mice were mated and housed in the institute’s animal care facility. Both males and females were used in equilibrated ratios. Bacteria from exponential growth cultures (OD_600_ = 1.0) were centrifuged and resuspended in sterile PBS at 1.67 × 10^8^ per mL for PAO1 and 3.33 × 10^8^ per mL for IHMA. Mice (8–10 weeks) were anesthetized by intraperitoneal administration of a mixture of xylazine (10 mg·Kg^−1^) and ketamine (50 mg·Kg^−1^). Then, 30 μL of bacterial suspension (5 × 10^6^ for PAO1 and 1 × 10^7^ for IHMA) were deposited in the animal’s nostrils.

### 4.17. Analysis of BALs

BALs were collected at 18 hpi by flushing the lungs of euthanized and intubated mice with 0.5 mL PBS (3 times) containing an anti-protease cocktail (Complete from Roche Diagnostics) and 5 mM EDTA. The collected fluids were centrifuged and both the pellets and the supernatants were snap-frozen and stored at −80 °C. Protein concentration in the supernatants was measured using the BCA kit from Thermo Scientific. IL-6 levels were measured using the mouse IL-6 ELISA kit from BioLegend. To evaluate the hemoglobin contents, pellets were thawed, resuspended in water and centrifuged. The supernatants were read at OD_560_. Dosages were performed once in triplicate.

### 4.18. Statistical Analyses

Statistical analyses were performed using GraphPad Prism software (v. 7.00). Normality was assessed using Shapiro–Wilk’s test. ExlA values measured in liquid cultures and BALs were analyzed by Kruskal–Wallis’s test, followed by Dunn’s post-hoc test. Linear regression curves and goodness-of-fit (*r*^2^) were calculated by the program. ExlA spot counts were analyzed with Chi2 test, followed for simple comparisons with the exact Fisher’s test. *p*-values > 0.5 were considered non-significant.

## Figures and Tables

**Figure 1 toxins-13-00645-f001:**
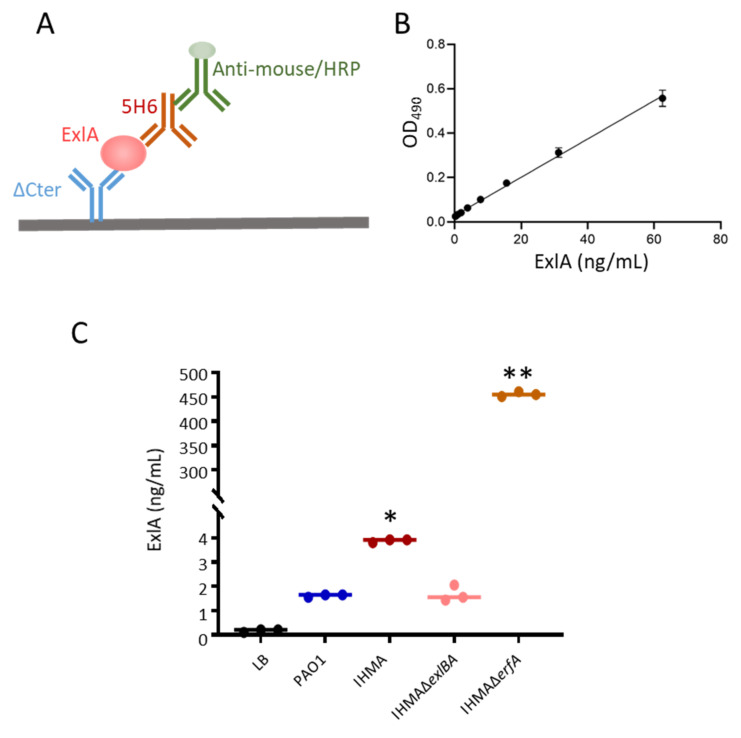
Quantification of ExlA in bacterial secretomes. (**A**). Scheme of the sandwich ELISA used for quantification of ExlA in liquid samples. The polyclonal antibody ΔCter and the monoclonal antibody 5H6 were used as capture and detection antibodies, respectively. (**B**). Reproducibility of the standard curve (mean +/− SD, *n* = 3 independent experiments) and range of the assay. (**C**). ExlA quantification in bacterial secretomes. Four strains were used: IHMA is a natural *exlA*-positive strain; IHMAΔ*exlBA* is an isogenic IHMA mutant deficient in ExlA secretion; IHMAΔ*erfA* is an isogenic mutant overproducing ExlA; PAO1 is an *exlA*-negative strain. LB medium was used as negative control. Secretomes were collected from cultures at OD_600_ 2.0 and supplemented with an anti-protease cocktail. All samples were assayed for ExlA content with the ExlA ELISA in triplicates. Individual values are shown together with the median (bar). Statistical differences were evaluated using Kruskal–Wallis’s test (*p* = 0.006), followed by Dunn’s test for comparison to control: * *p* = 0.03; ** *p* = 0.003; all other values were non-significant. The experiment was reproduced once with similar results.

**Figure 2 toxins-13-00645-f002:**
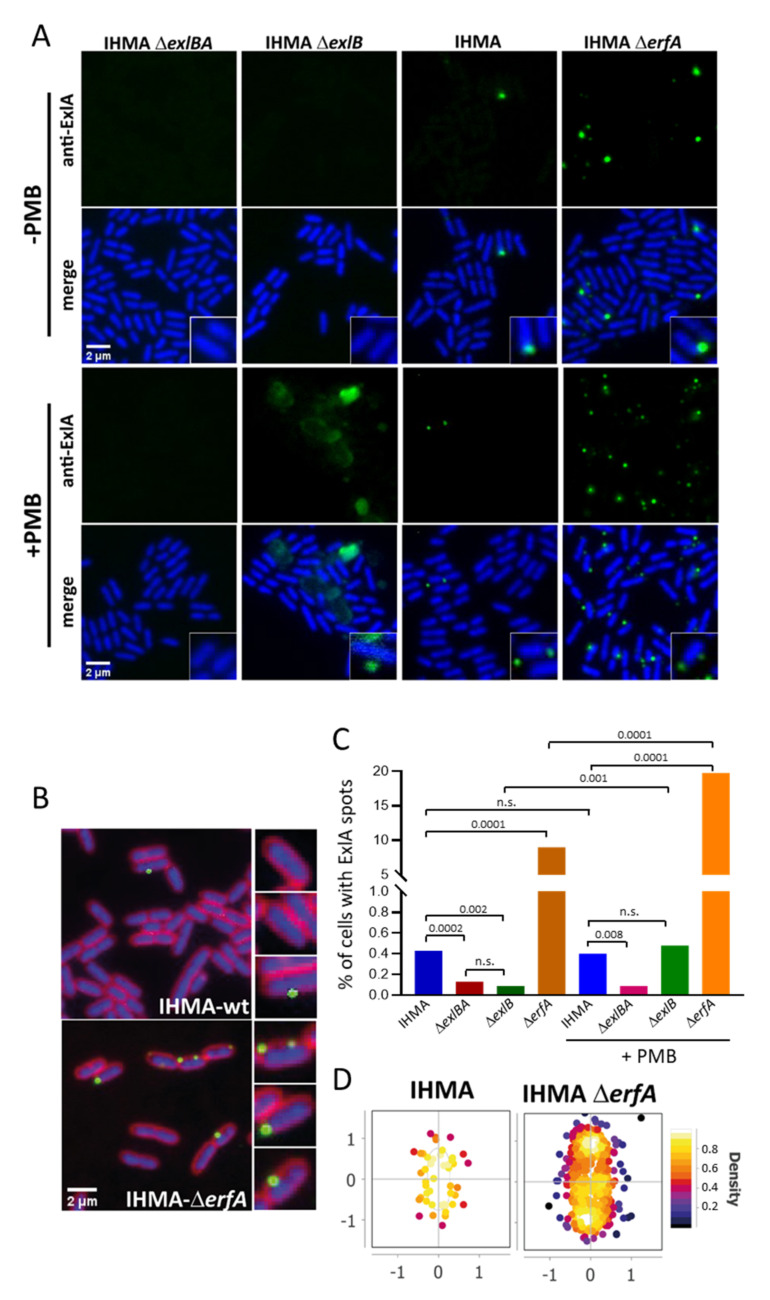
ExlA immunolocalization in bacteria. (**A**,**B**). Bacteria in exponential phase: IHMA, IHMA *exlB*-mut (IHMAΔ*exlB*), IHMAΔ*exlBA* and IHMAΔ*erfA* were fixed. Part of the bacteria was permeabilized with 100 µg/mL of polymyxin B (PMB), as indicated. Bacteria were then immunolabeled with monoclonal mouse anti-ExlA antibody (7G10) and subsequently with anti-mouse antibody coupled to Alexa-Fluor^®^ 488 (green). Hoechst labeling (blue) was used to stain the bacterial cytoplasm. FM4-64 staining was used in (**B**) for membrane labeling (red). (**A**). ExlA labeling and merge (ExlA + Hoechst) images are shown for all four strains, together with magnifications. (**B**). Merge immunolabeling images (ExlA + Hoechst + FM4-64) are shown for IHMA and IHMAΔ*erfA*. (**C**). The percentages of bacteria with ExlA spots were counted on 10–11 images per condition, using MicrobeJ software, and are shown as bars (*n* = 3302–11,465 cells analyzed in each condition). The Chi2 test was used to establish statistical differences (*p* < 0.0001), and dual comparisons were calculated using the two-sided Fisher’s exact test: *p*-values are indicated above the bars. n.s., non-significant. (**D**). Distribution of the ExlA spots on the bacterial circumference was determined using MicrobeJ. The heat map shows spot density associated with bacteria.

**Figure 3 toxins-13-00645-f003:**
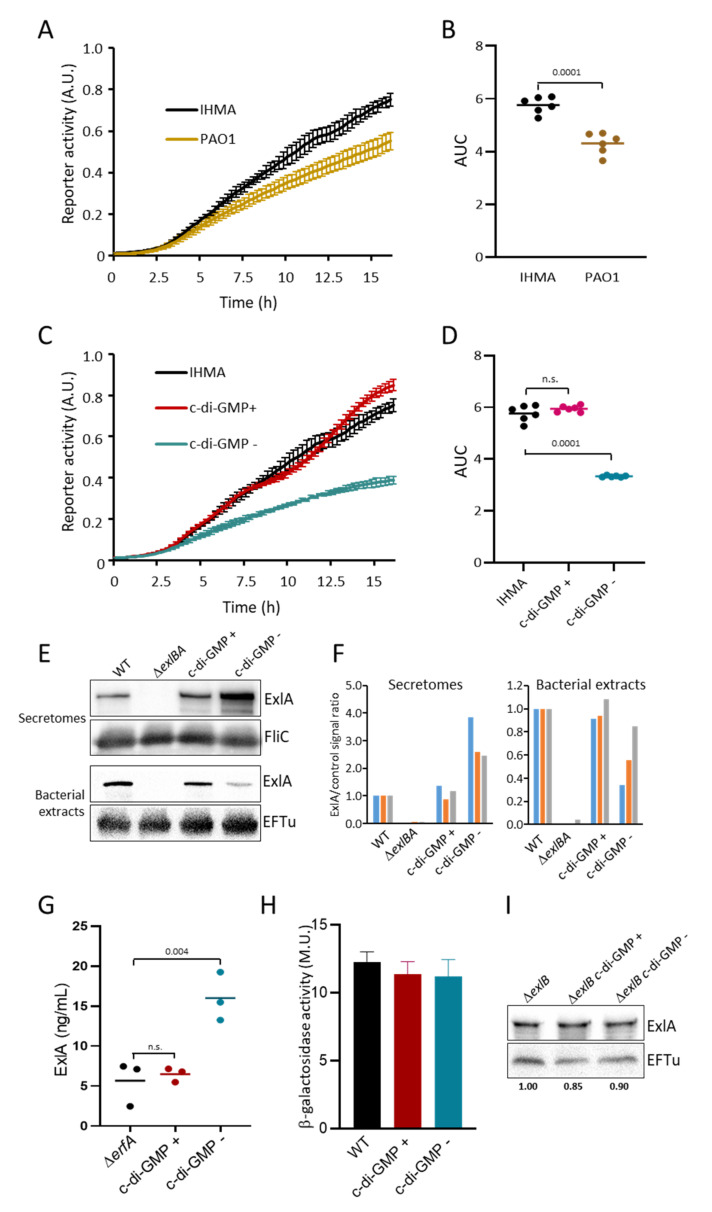
Cyclic-di-GMP regulates ExlA secretion(**A**). Intracellular c-di-GMP levels in IHMA::pSW196 (IHMA) and PAO1::pSW196 (PAO1). Both strains contained the chromosomal *PcdrA-gfp*(ASV)^c^ fusion as fluorescent reporter of c-di-GMP levels. GFP fluorescence was recorded for 16 h and was normalized by OD_585_ to assess bacterial growth. The results, in arbitrary units (A; U), represent the mean +/− SD of six replicates. (**B**). The areas under the curves (AUC) were deduced from data shown in (**A**). Bar: mean. The indicated *p*-value was calculated with the Student’s test. (**C**,**D**). Similar experiment using IHMA::pSW196 (IHMA), IHMA::pSW196-*wspR** (c-di-GMP +) and IHMA::pSW196-*PA2133* (c-di-GMP −). Gene expression was induced by arabinose 0.025%. Statistical differences between AUC were calculated with ANOVA (*p* < 0.0001) followed by Dunnett’s test for comparison with IHMA. (**E**). ExlA contents of secretomes and bacteria for IHMA::pSW196 (WT), IHMA Δ*exlBA*, IHMA::pSW196-*wspR** (c-di-GMP +) and IHMA::pSW196-*PA2133* (c-di-GMP −). Secretomes were concentrated 100X and bacterial extracts 10X before Western blot analysis and incubation with ExlA antibodies (Cter and ΔCter). FliC and EFTu were used as loading controls for secretomes and bacterial extracts, respectively. (**F**). ExlA/control signal ratios in secretomes and bacterial extracts are shown for three independent Western blot experiments (color coded). (**G**). ExlA was quantified by ELISA in the secretomes of IHMAΔ*erfA*::pSW196 (Δ*erfA*), IHMAΔ*erfA*::pSW196-*wspR** (c-di-GMP +) and IHMAΔ*erfA*::pSW196-*PA2133* (c-di-GMP −). Three clones were assayed in triplicates. The dots represent the data for each clone with the mean (bar). Global statistical difference was established with ANOVA (*p* = 0.0036) and *p*-values for individual comparisons with IHMAΔ*erfA* (Dunnett’s test) are shown. (**H**). The three IHMA derivatives described in (**C**,**D**) (WT, c-di-GMP + and c-di-GMP −) were analyzed in strains harboring *lacZ* integrated within *exlA* gene to measure the transcriptional activity of the *exlBA* promoter. The β-galactosidase activity was measured in triplicates when bacterial cultures reached OD_600_ = 1 and expressed in Miller’s units (MU). Results are shown as mean +/− SD. No significant differences were established with ANOVA. (**I**). ExlA contents in bacterial extracts (concentrated 10X) from strains IHMA*exlB*mut::pSW196 (Δ*exlB*), IHMA*exlB*mut::pSW196-*wspR** (Δ*exlB* c-di-GMP +), IHMA*exlBmut*::pSW196-*PA2133* (Δ*exlB* c-di-GMP −) were determined by Western blot. EFTu was used as loading control.

**Figure 4 toxins-13-00645-f004:**
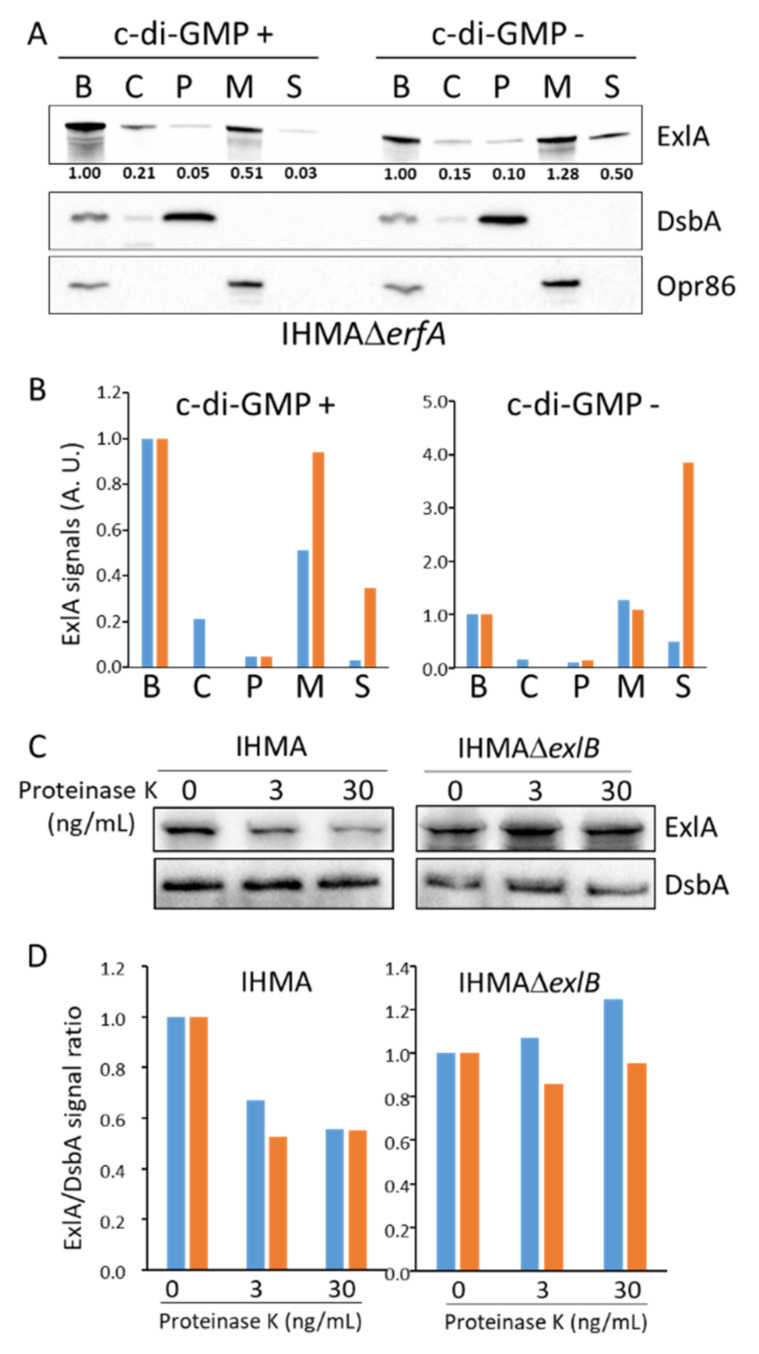
The c-di-GMP holds ExlA at the outer membrane. (**A**). Cellular fractionation of IHMA Δ*erfA* containing either pSW196-*wspR** (c-di-GMP +) or pSW196-*PA1233* (c-di-GMP −): B, total bacterial extract; C, cytosol; P, periplasm; M, membranes; S, secretome. Fractions were analyzed by Western blot to detect ExlA, as well as DsbA and Opr86, as markers for periplasm and membranes, respectively. (**B**). ExlA signal intensities are shown for two independent Western blot experiments (color coded) in arbitrary units. (**C**). Accessibility to proteinase K. Bacteria (IHMA or IHMA Δ*exlB*) were incubated 5 min on ice with increasing concentrations of proteinase K, and total extracts were analyzed by Western blot for detection of ExlA, as well as DsbA to control outer-membrane integrity. (**D**). ExlA/Dsba signal ratios are shown for two independent experiments (color coded).

**Figure 5 toxins-13-00645-f005:**
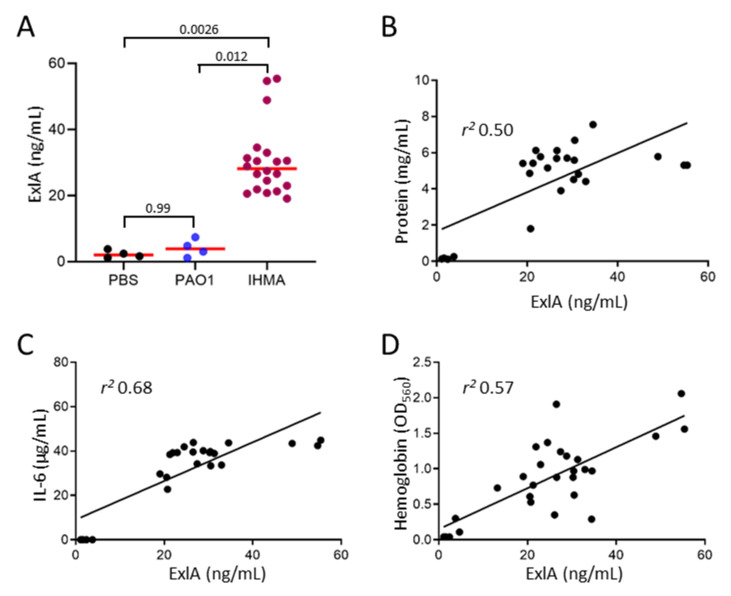
Quantification of ExlA in broncho-alveolar lavage fluids of infected mice. Pneumonia was induced in mice by inhalation of a suspension of IHMA (*exlA*+; *n* = 20) or PAO1 (*exlA*-; *n* = 4). In parallel, four mice were mock-infected with PBS. BAL fluids (1.5 mL of PBS with anti-proteases) were collected at 18 hpi and centrifuged. (**A**). ExlA concentration was measured on the supernatants. Individual values are shown together with the median. Data were analyzed by Kruskal–Wallis’s test (*p* = 0.0004). The *p*-value of multiple comparisons with Dunn’s test are shown. (**B**–**D**). Protein, IL-6 and hemoglobin data are shown in a correlation graph with ExlA concentrations. The linear regression curves are shown, as well as the *r*^2^ values. (**B**,**C**). Protein and IL-6 concentrations were measured in BAL supernatants. (**D**). Hemoglobin content was measured in BAL pellets and represented as OD_560_.

**Figure 6 toxins-13-00645-f006:**
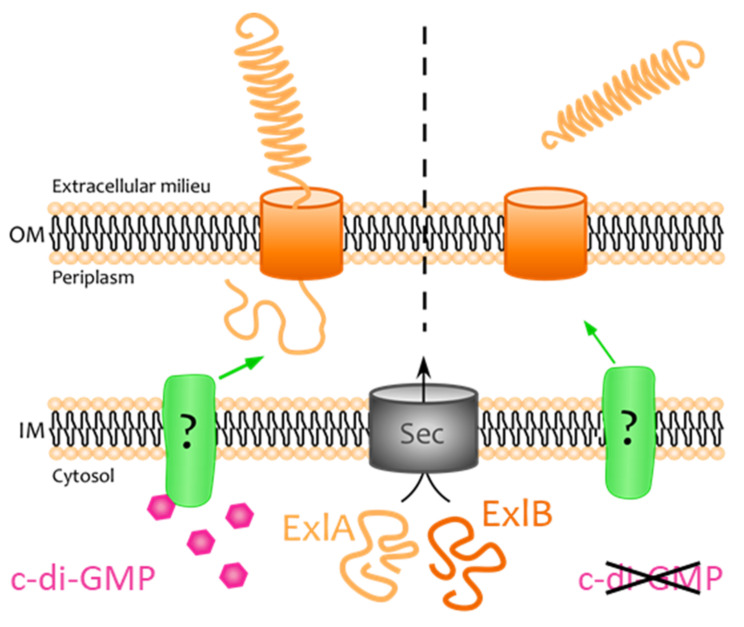
Proposed model for ExlA secretion regulation. ExlA and ExlB are transported into the periplasm by the Sec machinery. ExlB forms a pore inside the outer membrane, in which ExlA is maintained in conditions of high c-di-GMP levels and points outside. When c-di-GMP levels are low, ExlA is secreted into the extracellular milieu. The indirect pathway that conveys the cytosolic c-di-GMP signal to ExlA in the outer membrane is unknown.

**Table 1 toxins-13-00645-t001:** Bacterial strains and plasmids.

Names	Relevant Characteristics	Reference/Source
*E. coli* strain		
BL21 Star™ (DE3)	IPTG-inducible production of T7 RNA polymerase (prophage DE3), increased mRNA stability (*rne131*) and increased protein stability (lacks the Lon and OmpT proteases)	Life Technologies 3537305 10411735
*P. aeruginosa* strains		
IHMA879472 (“IHMA”)	*exlA* positive	[22]
IHMAΔ*exlBA*	IHMA isogenic mutant deleted in *exlB* and *exlA*	[15]
IHMAΔ*erfA*	IHMA isogenic mutant deficient in *erfA*	[15]
IHMA::pSW196	IHMA with empty pSW196 vector integrated at chromosomal *attB* site	This study
IHMA::pSW196-*wspR** (“c-di-GMP +”)	IHMA expressing WspR*	This study
IHMA::pSW196-*PA2133* (“c-di-GMP −”)	IHMA expressing PA2133	This study
IHMAΔ*erfA*::pSW196	IHMA isogenic mutant deficient in *erfA* with empty pSW196 vector integrated at chromosomal *attB* site	This study
IHMAΔ*erfA*::pSW196-wspR*	IHMA isogenic mutant deficient in *erfA* expressing WspR*	This Study
IHMAΔ*erfA*::pSW196-PA2133	IHMA isogenic mutant deficient in *erfA* expressing PA2133	This Study
IHMA::pSW196:Tn7-*pcdrA-gfp(ASV)^c^*	IHMA::pSW196 tagged with miniTn7 harboring *PcdrA-gfp* reporter fusion	This study
IHMA::pSW196-*wspR**: Tn7-*pcdrA-gfp(ASV)^c^*	IHMA-c-di-GMP + with chromosomal *PcdrA-gfp* fusion	This study
IHMA::pSW196-*PA2133*: Tn7-*pcdrA-gfp(ASV)^c^*	IHMA-c-di-GMP − with chromosomal *PcdrA-gfp* fusion	This study
IHMA *exlA::lacZ*	IHMA87 with promoterless *lacZ* in *exlA*	[15]
IHMA *exlA::lacZ*:pSW196	IHMA *exlA::lacZ* with integrated pSW196	This study
IHMA *exlA::lacZ*:pSW196-*wspR**	IHMA *exlA::lacZ expressing wspR**	This study
IHMA *exlA::lacZ*:pSW196-*PA2133*	IHMA *exlA::lacZ expressing PA2133*	This study
IHMA exlB-mut	IHMA with suicide pEXG2 plasmid inserted in *exlB*	[8]
IHMA*exlB-mut*::pSW196-*wspR**	IHMA *exlB*-mut expressing WspR*	This study
IHMA*exlB-mut*::pSW196-*PA2133*	IHMA *exlB*-mut expressing PA2133	This study
PAO1	*exlA* negative, T3SS positive	Reference strain
PAO1:: pSW196:Tn7-*pcdrA-gfp(ASV)^c^*	PAO1::pSW196 tagged with miniTn7 harboring *PcdrA*-*gfp* reporter fusion	This study
Bacterial plasmids		
pACYC-ompA-exlB	IPTG-inducible expression of codon-optimized *exlB* (*E. coli*) with signal peptide coding sequence replaced by that of *E. coli* ompA	This study
pET28-pelB-exlA	IPTG-inducible expression of codon-optimized *exlA* (*E. coli*) with signal peptide coding sequence replaced by that of *D. dadantii pelB*	[9]
pBBR1MCS4- R246A-wspR	Replicative pBBR1MCS-4 producing the highly active diguanylate cyclase WspR^R246A^ (WspR*)	[23]
pBBR1MCS4-PA2133	Replicative pBBR1MCS-4 producing the phosphodiesterase cyclase PA2133	[23]
pRK600	Helper plasmid with conjugative properties	[24]
pSW196	Site-specific integrative plasmid (*attB* site), *araC*- P*BAD* cassette	[25]
pSW196-*wspR**	pSW196 containing P*BAD* fused to *wspR** sequence subcloned from pBBR1MCS4-R246A-wspR	This study
pSW196-*PA2133*	pSW196 containing PB*AD* fused to *PA2133* sequence subcloned from pBBR1MCS4-PA2133	This study
pTn7 CdrA-gfp(ASV)*^c^*	Suicide vector for delivery of miniTn7 harboring *cdrA* promoter fused to unstable GFP-encoding sequence	[26]
pUX-BF13	Plasmid providing Tn*7* transposase genes in *trans*	[26]

## Data Availability

Not applicable.

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
