# Peer review of "ExlA Pore-Forming Toxin: Localization at the Bacterial Membrane, Regulation of Secretion by Cyclic-Di-GMP, and Detection In Vivo"

_toxins, 2021, doi:10.3390/toxins13090645_

Round 1

Reviewer 1 Report

The work of the authors presents novel insights into the secretion of the ExlA toxin of P. aeruginosa. ExlA is a pore-forming toxin responsible for the virulence of T3SS-deficient P. aeruginosa isolates. We currently lack an understanding of ExlA secretion from bacteria and its insertion into the host cell membrane. The host cell toxicity of ExlA is contact-dependent, and it is assumed that ExlA is entrapped within bacteria and delivered only upon interaction with a receptor located on the target cell.   

The authors use immunofluorescence labeling to show that ExlA accumulates at specific locations of the bacterial membrane and this localization depends on ExlB transporter protein. Accordingly, the accessibility of ExlA to proteinase K also depends on ExlB. This is the major finding of the paper.

The authors also show that ExlA is poorly secreted in liquid culture while being highly detectable in BAL lavages of the infected mice. However, this might be caused by the killing of bacteria and/or detection of ExlA from the host cell, and not related to an increase in the secretion of ExlA per se.    

My major issue with the work is the data showing the role of cyclic-di-GMP in ExlA secretion. The effect seems to be minor and is demonstrated by western blots, which lack quantification (Fig. 3E, and Fig. 4A). Further, an indication of the number of repetitions is missing. Moreover, the majority of the ExlA molecules still remain entrapped in the bacterial outer membrane in c-di-GMP- strain in Fig.4A. Could authors quantify the level of ExlA secretion in c-di-GMP- conditions by ELISA and add Western blot quantification?

Author Response

Reviewer 1

The work of the authors presents novel insights into the secretion of the ExlA toxin of P. aeruginosa. ExlA is a pore-forming toxin responsible for the virulence of T3SS-deficient P. aeruginosa isolates. We currently lack an understanding of ExlA secretion from bacteria and its insertion into the host cell membrane. The host cell toxicity of ExlA is contact-dependent, and it is assumed that ExlA is entrapped within bacteria and delivered only upon interaction with a receptor located on the target cell.   

The authors use immunofluorescence labeling to show that ExlA accumulates at specific locations of the bacterial membrane and this localization depends on ExlB transporter protein. Accordingly, the accessibility of ExlA to proteinase K also depends on ExlB. This is the major finding of the paper.

The authors also show that ExlA is poorly secreted in liquid culture while being highly detectable in BAL lavages of the infected mice. However, this might be caused by the killing of bacteria and/or detection of ExlA from the host cell, and not related to an increase in the secretion of ExlA per se.   

We agree. We included this possibility in the revised version.

My major issue with the work is the data showing the role of cyclic-di-GMP in ExlA secretion. The effect seems to be minor and is demonstrated by western blots, which lack quantification (Fig. 3E, and Fig. 4A). Further, an indication of the number of repetitions is missing. Moreover, the majority of the ExlA molecules still remain entrapped in the bacterial outer membrane in c-di-GMP- strain in Fig.4A. Could authors quantify the level of ExlA secretion in c-di-GMP- conditions by ELISA and add Western blot quantification?

It is not an “on-off” effect but it is still clear. The overproduction of PA2133 reduces the global intracellular c-di-GMP level. However, it is know that c-di-GMP action is often local, as the amounts of EAL and PDE enzymes are high in bacteria. We can infere or hypothesize that a dedicated EAL controls ExlA membrane anchoring.

We added Western blot quantifications in the figures. We believe that it is the best technique to observe the amounts of ExlA in this experiment, because 1/ it is possible to measure ExlA in the different compartments simultanously, and 2/ ExlA concentration in IHMA secretomes is very low and too close to the background to observe a clear effect.

Reviewer 2 Report

I have read the manuscript. The experiments themselves are generally fine (some minor issues are described later). However, I have some major concerns with the text itself, which I would like to share with you.

First, the title of the manuscript is “ExlA pore-forming toxin: localization at the bacterial membrane, mechanism of secretion regulation and detection in vivo”. Unfortunately, I did not see even one evidence for regulation of ExlA secretion, not to mention a mechanism of that regulation.

The Results section starts with a description of a development of a sandwich ELISA, the reasoning for developing that assay is not clear until the Discussion. The authors should explain that in detail. This part could be moved to the Supplementary Information, or at least moved to the end of the Results section.

The authors then discuss ExlA localization at the bacterial membrane which is interesting although not surprising given previous findings. Co-localization with ExlB could contribute to support the proposed mechanism of secretion.

Then, the real problems arise. The authors note that “IHMA bacteria tend to form small aggregates and are hyperadherent to plasticware (not shown), a phenotype known to be activated by the second messenger c-di-GMP. Furthermore, IHMA exhibits a reduced swarming motility, in agreement with a high cellular content of c-di-GMP” (page 5, lines 146-149). From there, they proceed to show intracellular levels of c-di-GMP. They do not show that directly but instead they use the PcdrA-gfp reporter system, where the cdrA promoter is activated by c-di-GMP. Figure 3A and 3B show that reporter activity increases for both the IHMA and PAO1 strains. From that, they deduce that “IHMA contains high amounts of c-di-GMP”. This sentence does not reflect properly the experimental result. An accurate sentence would be “IHMA shows higher GFP reporter activity compared to PAO1”. However, saying this sentence would not support their hypothesis that c-di-GMP levels in IHMA are high. Given that these are the days of EURO 2020, I would say that this deserves a “yellow card”.

The authors explain they “reasoned that if c-di-GMP regulates ExlA synthesis or secretion, manipulating c-di-GMP concentrations should alter toxin release.” (page 6, lines 156-157). However, manipulating the second messenger c-di-GMP concentrations alter many other things as well. How can you tell if c-di-GMP is directly linked to ExlA or only indirectly? They authors manage to modulate the c-di-GMP levels using either wspR* for increased expression and PA2133 for reduced expression. They then see that there is no significant difference in GFP reporter activity between IHMA and IHMA expressing wspR*. According to the authors, the finding is “possibly indicating that c-di-GMP concentrations are already high in wild-type strains” (page 6, lines 163-164). It could be the case or not, there is no way to conclude that based on current experimental results.

Anyway, they show higher ExlA levels in secretomes of IHMA expressing PA2133 compared to IHMA and IHMA expressing wspR*. In correspondence, they show lower ExlA levels in bacterial extractes of IHMA expressing PA2133 compared to IHMA and IHMA expressing wspR*. This is nice and interesting. However, it is misleading to claim that low c-di-GMP levels regulate ExlA levels. The claim, “lowering c-di-GMP increases ExlA release” (page 6, lines 169-170) is just misleading. Sadly, this deserves a “red card”.

I would like to add that the rationale behind cyclic-di-GMP controlling ExlA secretion is not clear. On one side, IHMA is known to have high levels of cyclic-di-GMP, on the other hand, it seems that low levels are required for secretion of ExlA. This is not clear.

I am sorry for this negative review, however my opinion is that this paper should be re-written and much more emphasis is put on adherence of the text to the actual results.

Author Response

Reviewer 2

I have read the manuscript. The experiments themselves are generally fine (some minor issues are described later). However, I have some major concerns with the text itself, which I would like to share with you.

First, the title of the manuscript is “ExlA pore-forming toxin: localization at the bacterial membrane, mechanism of secretion regulation and detection in vivo”. Unfortunately, I did not see even one evidence for regulation of ExlA secretion, not to mention a mechanism of that regulation.

c-di-GMP is a major regulator of P. aeruginosa behavior. In this paper, we show that alteration of c-di-GMP intracellular level modifies ExlA release. We thus conclude, rightly, that c-di-GMP regulates ExlA secretion. We agree with Reviewer 2 when he mentions that we did not determine the regulation mechanism by c-di-GMP. We thus modified the title accordingly.

The Results section starts with a description of a development of a sandwich ELISA, the reasoning for developing that assay is not clear until the Discussion. The authors should explain that in detail. This part could be moved to the Supplementary Information, or at least moved to the end of the Results section.

The reasoning of ExlA ELISA development is now better explained. We decided to maintain the ELISA description, although technical, in the Results section, because being able to quantify ExlA is an major advance of this study that can be useful for ExlA measurement in patients. The ELISA description must be at the beginning of the Results, as it is used in Fig. 1 for ExlA detection in bacterial secretomes.

The authors then discuss ExlA localization at the bacterial membrane which is interesting although not surprising given previous findings. Co-localization with ExlB could contribute to support the proposed mechanism of secretion.

In previous work, we showed that type 4 pili were required for optimal ExlA toxicity and that secretomes were not toxic. Several hypotheses could be drawn from these data. Herein, we show that ExlA is located at the bacterial membrane. This was not “surprising”, but remains an important finding.

Unfortunately, there is no anti-ExlB antibody available for colocalization experiments.

Then, the real problems arise. The authors note that “IHMA bacteria tend to form small aggregates and are hyperadherent to plasticware (not shown), a phenotype known to be activated by the second messenger c-di-GMP. Furthermore, IHMA exhibits a reduced swarming motility, in agreement with a high cellular content of c-di-GMP” (page 5, lines 146-149). From there, they proceed to show intracellular levels of c-di-GMP. They do not show that directly but instead they use the PcdrA-gfp reporter system, where the cdrA promoter is activated by c-di-GMP. Figure 3A and 3B show that reporter activity increases for both the IHMA and PAO1 strains. From that, they deduce that “IHMA contains high amounts of c-di-GMP”. This sentence does not reflect properly the experimental result. An accurate sentence would be “IHMA shows higher GFP reporter activity compared to PAO1”. However, saying this sentence would not support their hypothesis that c-di-GMP levels in IHMA are high. Given that these are the days of EURO 2020, I would say that this deserves a “yellow card”.

This is correct. We now conclude: “IHMA contains higher c-di-GMP amounts than PAO1”.

The authors explain they “reasoned that if c-di-GMP regulates ExlA synthesis or secretion, manipulating c-di-GMP concentrations should alter toxin release.” (page 6, lines 156-157). However, manipulating the second messenger c-di-GMP concentrations alter many other things as well. How can you tell if c-di-GMP is directly linked to ExlA or only indirectly?

c-di-GMP is indeed a large-spectrum regulator, but we do not pretend that it is a direct effect. In fact, it cannot be as c-di-GMP is not present in the periplasm, as shown in the proposed model.

They authors manage to modulate the c-di-GMP levels using either wspR* for increased expression and PA2133 for reduced expression. They then see that there is no significant difference in GFP reporter activity between IHMA and IHMA expressing wspR*. According to the authors, the finding is “possibly indicating that c-di-GMP concentrations are already high in wild-type strains” (page 6, lines 163-164). It could be the case or not, there is no way to conclude that based on current experimental results.

We agree. This is why we wrote “possibly”.

Anyway, they show higher ExlA levels in secretomes of IHMA expressing PA2133 compared to IHMA and IHMA expressing wspR*. In correspondence, they show lower ExlA levels in bacterial extractes of IHMA expressing PA2133 compared to IHMA and IHMA expressing wspR*. This is nice and interesting. However, it is misleading to claim that low c-di-GMP levels regulate ExlA levels. The claim, “lowering c-di-GMP increases ExlA release” (page 6, lines 169-170) is just misleading. Sadly, this deserves a “red card”.

We do not agree with Reviewer 2’s comment, as it is exactly what we observed: lowering c-di-GMP levels increases ExlA release, even if c-di-GMP acts in an indirect manner on this phenomenon.

I would like to add that the rationale behind cyclic-di-GMP controlling ExlA secretion is not clear. On one side, IHMA is known to have high levels of cyclic-di-GMP, on the other hand, it seems that low levels are required for secretion of ExlA. This is not clear.

If ExlA has to be anchored at the membrane to be active, it means that c-di-GMP enhances bacterial toxicity.On the contrary, low c-di-GMP levels induce ExlA release and a decrease of toxicity. This is indeed what we observed in toxicity assays, but we did not include these results in the article, as c-di-GMP may have several different effects in bacteria, which may influence bacterial toxicity independently of ExlA release.

I am sorry for this negative review, however my opinion is that this paper should be re-written and much more emphasis is put on adherence of the text to the actual results.

We hope that we answered to Reviewer 2’s concerns.

Reviewer 3 Report

This work is intresting and well-written. Τhe results are presented in detail with a satisfactory explanation.

Author Response

Reviewer 3

This work is intresting and well-written. Τhe results are presented in detail with a satisfactory explanation.

We thank Reviewer 3 for his very positive comments.

Reviewer 4 Report

Exolysin (ExlA) is a pore-forming toxin (PFT) secreted by P. aeruginosa which is a major opportunistic pathogen causing nosocomial infections. ExlA exerts its cytotoxic effect by inserting a pore of diameter ~1.6 nm of diameter and triggering both Ca2+ influx and K+ efflux. This PFT is part of a two-partner secretion system where ExlA is the passenger protein while ExlB is the bacterial membrane located transporter. The paper here describes the localization of ExlA at the bacterial membrane which is further regulated by cyclic-di-GMP (c-di-GMP) amount. The lower c-di-GMP level secretes ExlA via ExlB, whereas the higher-level retain the toxin at the membrane. The author also shows the ExlA is highly detected in broncho-alveolar lavage fluids of mice exposed to exlA+ strain and also developed an immunological test to quantify ExlA in biological samples.

Here are my few comments on the paper:

  • The paper showed the higher secretion of ElxA in broncho-alveolar lavage fluids of mice infected with an exlA+ strain. However, the toxin was poorly secreted in liquid culture. Therefore, what would be the justification of the study conducted in this study pertaining to membrane localization of ExlA? Does membrane localization occur only in liquid culture and not in the in-vivo model? Does the author try to attempt some experiments (like fluorescence) on lungs cells (from mice infected with the strain) to detect membrane localization?
  • It was fascinating to see ExlA accumulate at specific locations of the bacterial membrane. To my opinion, it will also be interesting to see if ExlB does the same.
  • There are western blots images (Fig 3E and Fig 4), I would suggest quantifying the bands.
  • Figure 2C, the percentages of bacteria with ExlA spots are shown. Would it be possible to include the number of cells counted, maybe the range (N = Lower number – higher number).
  • Minor edits like line 529 , 4.15 remove dot β-. galactosidase activity.

Author Response

Reviewer 4

Exolysin (ExlA) is a pore-forming toxin (PFT) secreted by P. aeruginosa which is a major opportunistic pathogen causing nosocomial infections. ExlA exerts its cytotoxic effect by inserting a pore of diameter ~1.6 nm of diameter and triggering both Ca2+ influx and K+ efflux. This PFT is part of a two-partner secretion system where ExlA is the passenger protein while ExlB is the bacterial membrane located transporter. The paper here describes the localization of ExlA at the bacterial membrane which is further regulated by cyclic-di-GMP (c-di-GMP) amount. The lower c-di-GMP level secretes ExlA via ExlB, whereas the higher-level retain the toxin at the membrane. The author also shows the ExlA is highly detected in broncho-alveolar lavage fluids of mice exposed to exlA+ strain and also developed an immunological test to quantify ExlA in biological samples.

Here are my few comments on the paper:

  • The paper showed the higher secretion of ElxA in broncho-alveolar lavage fluids of mice infected with an exlA+ strain. However, the toxin was poorly secreted in liquid culture. Therefore, what would be the justification of the study conducted in this study pertaining to membrane localization of ExlA? Does membrane localization occur only in liquid culture and not in the in-vivo model? Does the author try to attempt some experiments (like fluorescence) on lungs cells (from mice infected with the strain) to detect membrane localization?

This difference between in vitro and in vivo experiments is indeed intriguing. All hypotheses are open. We tried to detect ExlA with 7G10 antibody (the only one that yield a specific labeling on bacteria) on tissue sections of infected lungs. As 7G10 is a mouse monoclonal, we first blocked the mouse immunoglobulins in tissues, but unfortunately, the background was still too high to detect a specific signal. We tried by immunofluorescence and immunochemistry with no success. As suggested by Reviewer 1, the detected ExlA in BALs may also come from cellular debris (from mouse cells or bacteria) that are not eliminated by centrifugation. We now include this hypothesis in the revised version. However, we consider that the main result of the BAL data is the fact that ExlA is detectable in vivo, and correlates with disease severity.

  • It was fascinating to see ExlA accumulate at specific locations of the bacterial membrane. To my opinion, it will also be interesting to see if ExlB does the same.

There is no anti-ExlB antibody available, but all our results tend to show that ExlA is linked to ExlB at the membrane.

  • There are western blots images (Fig 3E and Fig 4), I would suggest quantifying the bands.

We quantified the bands and show the results on the figures.

  • Figure 2C, the percentages of bacteria with ExlA spots are shown. Would it be possible to include the number of cells counted, maybe the range (N = Lower number – higher number).

This is now included.

  • Minor edits like line 529 , 4.15 remove dot β-. galactosidase activity.

This error has been corrected.

Round 2

Reviewer 1 Report

The effect of cyclic-di-GMP in ExlA secretion is still not convincingly demonstrated. Authors should add ELISA quantification of the level of ExlA secretion in their c-di-GMP- conditions.

Why to develop the ELISA method and speak of it very highly compared to WB (lines 77-80), and then NOT TO USE IT in supporting the claims? Moreover, authors added quantification of a single WB into their manuscript, claiming it is representative of 3 and 2 experiments, respectively.  No means +/- s.d. are shown.

Author Response

Reviewer 1

The effect of cyclic-di-GMP in ExlA secretion is still not convincingly demonstrated. Authors should add ELISA quantification of the level of ExlA secretion in their c-di-GMP- conditions.

Why to develop the ELISA method and speak of it very highly compared to WB (lines 77-80), and then NOT TO USE IT in supporting the claims?

As mentioned previously, the ELISA values for IHMA secretomes are too close to the background level and not adapted to demonstrate the effect of c-di-GMP downregulation. To overcome this issue, we performed ELISA assays in IHMAΔerfA background. The data (Fig. 3G) clearly show that lowering c-di-GMP increases ExlA contents in secretomes.

Moreover, authors added quantification of a single WB into their manuscript, claiming it is representative of 3 and 2 experiments, respectively.  No means +/- s.d. are shown.

We now include histograms for Western blot quantifications (Fig. 3F, 4B, 4D). We believe that it is better than means +/- sd, as the color code allows the identification of each experiment in each condition.

We hope we convinced Reviewer 1 that lowering c-di-GMP increases ExlA release.

Reviewer 2 Report

Reviewer 2

I have read the manuscript. The experiments themselves are generally fine (some minor issues are described later). However, I have some major concerns with the text itself, which I would like to share with you.

First, the title of the manuscript is “ExlA pore-forming toxin: localization at the bacterial membrane, mechanism of secretion regulation and detection in vivo”. Unfortunately, I did not see even one evidence for regulation of ExlA secretion, not to mention a mechanism of that regulation.

c-di-GMP is a major regulator of P. aeruginosa behavior. In this paper, we show that alteration of c-di-GMP intracellular level modifies ExlA release. We thus conclude, rightly, that c-di-GMP regulates ExlA secretion. We agree with Reviewer 2 when he mentions that we did not determine the regulation mechanism by c-di-GMP. We thus modified the title accordingly.

IMO, the word “mechanism” should not appear in the title.

IMO, indirect regulation of ExlA secretion by c-di-GMP is not the same as direct regulation.

Please note that Supp Information title is the old title.

The Results section starts with a description of a development of a sandwich ELISA, the reasoning for developing that assay is not clear until the Discussion. The authors should explain that in detail. This part could be moved to the Supplementary Information, or at least moved to the end of the Results section.

The reasoning of ExlA ELISA development is now better explained. We decided to maintain the ELISA description, although technical, in the Results section, because being able to quantify ExlA is an major advance of this study that can be useful for ExlA measurement in patients. The ELISA description must be at the beginning of the Results, as it is used in Fig. 1 for ExlA detection in bacterial secretomes.

OK.

The authors then discuss ExlA localization at the bacterial membrane which is interesting although not surprising given previous findings. Co-localization with ExlB could contribute to support the proposed mechanism of secretion.

In previous work, we showed that type 4 pili were required for optimal ExlA toxicity and that secretomes were not toxic. Several hypotheses could be drawn from these data. Herein, we show that ExlA is located at the bacterial membrane. This was not “surprising”, but remains an important finding.

Unfortunately, there is no anti-ExlB antibody available for colocalization experiments.

OK.

Then, the real problems arise. The authors note that “IHMA bacteria tend to form small aggregates and are hyperadherent to plasticware (not shown), a phenotype known to be activated by the second messenger c-di-GMP. Furthermore, IHMA exhibits a reduced swarming motility, in agreement with a high cellular content of c-di-GMP” (page 5, lines 146-149). From there, they proceed to show intracellular levels of c-di-GMP. They do not show that directly but instead they use the PcdrA-gfp reporter system, where the cdrA promoter is activated by c-di-GMP. Figure 3A and 3B show that reporter activity increases for both the IHMA and PAO1 strains. From that, they deduce that “IHMA contains high amounts of c-di-GMP”. This sentence does not reflect properly the experimental result. An accurate sentence would be “IHMA shows higher GFP reporter activity compared to PAO1”. However, saying this sentence would not support their hypothesis that c-di-GMP levels in IHMA are high. Given that these are the days of EURO 2020, I would say that this deserves a “yellow card”.

This is correct. We now conclude: “IHMA contains higher c-di-GMP amounts than PAO1”.

OK.

The authors explain they “reasoned that if c-di-GMP regulates ExlA synthesis or secretion, manipulating c-di-GMP concentrations should alter toxin release.” (page 6, lines 156-157). However, manipulating the second messenger c-di-GMP concentrations alter many other things as well. How can you tell if c-di-GMP is directly linked to ExlA or only indirectly?

c-di-GMP is indeed a large-spectrum regulator, but we do not pretend that it is a direct effect. In fact, it cannot be as c-di-GMP is not present in the periplasm, as shown in the proposed model.

It should be clearly noted in the text that the expected effect is indirect. The word “indirect” should appear in the text. Currently, it does not appear, not even once.

They authors manage to modulate the c-di-GMP levels using either wspR* for increased expression and PA2133 for reduced expression. They then see that there is no significant difference in GFP reporter activity between IHMA and IHMA expressing wspR*. According to the authors, the finding is “possibly indicating that c-di-GMP concentrations are already high in wild-type strains” (page 6, lines 163-164). It could be the case or not, there is no way to conclude that based on current experimental results.

We agree. This is why we wrote “possibly”.

IMO, “possibly” is inappropriate here. This sentence should be revised. As I wrote earlier, it could be the case or not. Please stick to conclusions that are based on experimental results.

Anyway, they show higher ExlA levels in secretomes of IHMA expressing PA2133 compared to IHMA and IHMA expressing wspR*. In correspondence, they show lower ExlA levels in bacterial extractes of IHMA expressing PA2133 compared to IHMA and IHMA expressing wspR*. This is nice and interesting. However, it is misleading to claim that low c-di-GMP levels regulate ExlA levels. The claim, “lowering c-di-GMP increases ExlA release” (page 6, lines 169-170) is just misleading. Sadly, this deserves a “red card”.

We do not agree with Reviewer 2’s comment, as it is exactly what we observed: lowering c-di-GMP levels increases ExlA release, even if c-di-GMP acts in an indirect manner on this phenomenon.

Again, IMO, the conclusions of the authors are not based on their experimental results. What we see is that in cells with lower amounts of c-di-GMP (actually, cells with lower GFP reporter activity), ExlA release is increased. The authors did not supply sufficient evidence to support their idea that lowering c-di-GMP increases ExlA release. The authors should revise their text to reflect that or add appropriate experiments.

I would like to add that the rationale behind cyclic-di-GMP controlling ExlA secretion is not clear. On one side, IHMA is known to have high levels of cyclic-di-GMP, on the other hand, it seems that low levels are required for secretion of ExlA. This is not clear.

If ExlA has to be anchored at the membrane to be active, it means that c-di-GMP enhances bacterial toxicity.On the contrary, low c-di-GMP levels induce ExlA release and a decrease of toxicity. This is indeed what we observed in toxicity assays, but we did not include these results in the article, as c-di-GMP may have several different effects in bacteria, which may influence bacterial toxicity independently of ExlA release.

Maybe I missed something? I thought IHMA shows both high levels of cyclic-di-GMP and high ExlA release. 

I am sorry for this negative review, however my opinion is that this paper should be re-written and much more emphasis is put on adherence of the text to the actual results.

We hope that we answered to Reviewer 2’s concerns.

Author Response

Reviewer 2

IMO, the word “mechanism” should not appear in the title.

The word “mechanism” had been removed from the title.

IMO, indirect regulation of ExlA secretion by c-di-GMP is not the same as direct regulation.

We agree. This point has been clarified in the text (see below).

Please note that Supp Information title is the old title.

This was a mistake. The title has been corrected.

It should be clearly noted in the text that the expected effect is indirect. The word “indirect” should appear in the text. Currently, it does not appear, not even once.

The fact that the regulation is indirect is now inserted in the Discussion and Fig. 6 legend.

They authors manage to modulate the c-di-GMP levels using either wspR* for increased expression and PA2133 for reduced expression. They then see that there is no significant difference in GFP reporter activity between IHMA and IHMA expressing wspR*. According to the authors, the finding is “possibly indicating that c-di-GMP concentrations are already high in wild-type strains” (page 6, lines 163-164). It could be the case or not, there is no way to conclude that based on current experimental results.

IMO, “possibly” is inappropriate here. This sentence should be revised. As I wrote earlier, it could be the case or not. Please stick to conclusions that are based on experimental results.

We modified the sentence to remove this hypothesis.

Again, IMO, the conclusions of the authors are not based on their experimental results. What we see is that in cells with lower amounts of c-di-GMP (actually, cells with lower GFP reporter activity), ExlA release is increased. The authors did not supply sufficient evidence to support their idea that lowering c-di-GMP increases ExlA release. The authors should revise their text to reflect that or add appropriate experiments.

In this experiment, we overexpressed a phosphodiesterase known to decrease c-di-GMP levels, and we verified its effect using a published (Ref. 23) and valid method to monitor c-di-GMP concentration (Fig 3C,D). We concluded that c-di-GMP level was indeed decreased when the phosphodiesterase was expressed. Furthermore, we observed that ExlA release was increased when c-di-GMP is lowered (Fig. 3E). Therefore, we concluded that lowering c-di-GMP increase ExlA release, and now insist on the fact that this effect is indirect, as c-di-GMP is not present in the periplasm. We believe that this conclusion is justified.

Maybe I missed something? I thought IHMA shows both high levels of cyclic-di-GMP and high ExlA release. 

When we measured ExlA concentration in IHMA secretomes (Fig. 1C), we observed “little but significant” ExlA levels, just above the background, i.e., as measured in IHMAΔexlBA or PAO1 secretomes. As IHMA is one of best ExlA producer (as detected by Western blot; in ref. 6), we thus concluded in the same paragraph: “This feature confirms that ExlA is poorly secreted from wild-type exlA+ strains in these condtions…”.

Round 3

Reviewer 1 Report

The reviewer would like to thank the authors for a robust demonstration of the effects of the c-di-GMP. 

Reviewer 2 Report

The authors answered all my questions and made all required corrections to the manuscript.